# Mitigating World Biases: A Multimodal Multi-View Debiasing Framework for Fake News Video Detection

## ABSTRACT

Short videos turn into an important channel for public sharing, as well as they've become a fertile ground for fake news. Fake news video detection is to judge the veracity of news based on its different modal information, such as video, audio, text, image and social context information. Current detection models tend to learn the multimodal dataset biases within spurious correlations between news modalities and veracity labels as shortcuts, rather than learning how to integrate the multimodal information behind them to reason, resulting in seriously degrading their detection and generalization capabilities. To address this issues, we propose a Multimodal Multi-View Debiasing (MMVD) framework, which makes the first attempt to mitigate various multimodal biases for fake news video detection. Inspired by people's misleading situations by multimodal short videos, we summarize three cognitive biases: static, dynamic and social biases. MMVD put forward a multi-view causal reasoning strategy to learn unbiased dependencies within the cognitive biases, thus enhancing the unbiased prediction of multimodal videos. The extensive experimental results show that the MMVD could improve the detection performance of multimodal fake news video. Studies also confirm that our MMVD can mitigate multiple biases on complex real-world scenarios and improve generalization ability of fake news video detection.

## CCS CONCEPTS

• **Information systems** → **Multimedia information systems**; *Social networks.*

## KEYWORDS

Fake news video detection, Multi-view, Debiasing

## 1 INTRODUCTION

With the evolution ways of sharing news on social media, short videos become a popular channel for news dissemination [31]. On short video platforms, individuals can effectively communicate news, opinions and emotions through multimodal information. They utilize a variety of mediums including text, images, videos and audio to engage deeply in discussions about current events and topics of interest [19]. However, without professional means of debunking the news, short video news is far easier to widespread and

**Unpublished working draft. Not for distribution.**

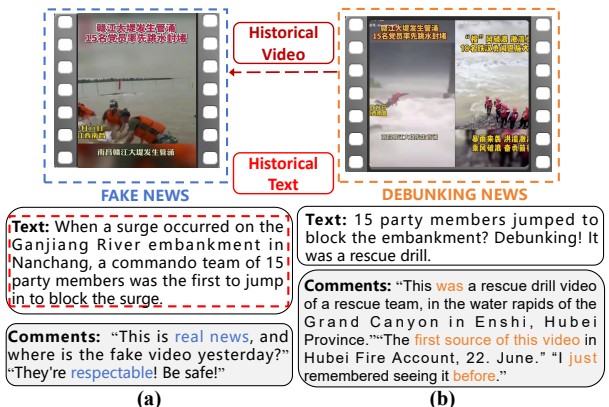

**Figure 1: Typical cases of bias in news videos. (a) shows a fake news video, which uses the historical authoritative video, text and misleading comments to deceive public. (b) shows a debunking news video containing debunking news video contents and comments.**

rapid spread on social platforms, especially during periods of international political events and public health events [28]. During the initial stages of the Ukraine war, a TikTok content creator garnered attention from over 30 million users by fabricating a sensationalized depiction of the conflict [26]. Similarly, amid the explosive spread of the Covid-19 pandemic, purveyors of misinformation on social media produced deceptive video content, hindering efforts in epidemic prevention [22]. Consequently, the detection of fake news videos on social media platforms becomes increasingly imperative.

Traditional fake news video detection methods applied machine learning models to detect fake news by utilizing contextual features from video titles and comments [18, 22]. With the advancement of deep learning, several fake news detectors have integrated extra features such as video frameworks [3], visual-speech [23], social context [19], as well as cross-modal consistency features [13], to jointly train more effective multimodal features for fake news video detection. These existing studies have primarily concentrated on leveraging provided news contents to capture label-specific features for unknown news. However, these methodologies may introduce various biases, such as textual bias [21], visual bias [2], video bias [14] and social bias [37]. For instance, as depicted in Figure 1(a), a short video reporting a positive event resonated with the public. Figure 1(b) shows a debunking video argued that this fake video in Figure 1(a) was manipulated by incorporating historical authoritative footage, textual content and misleading comments to deceive the public [19]. Such biases stemming from historical and misleading multimodal information can substantially compromise a model's detection and generalization capabilities, rendering it susceptible to adversarial attacks [8].

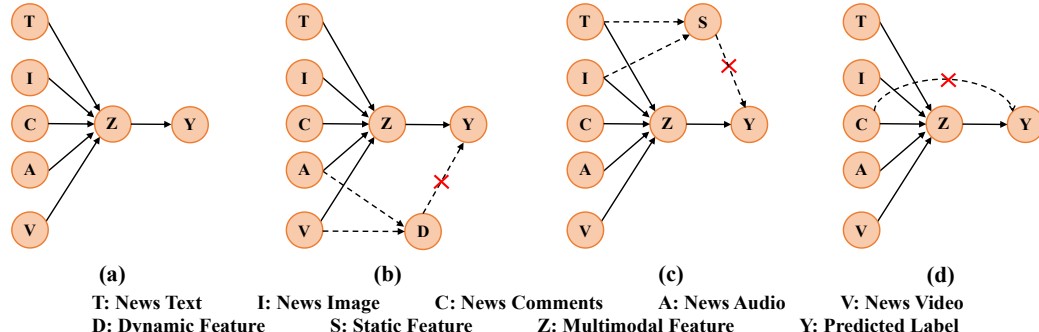

**(a)**     **(b)**     **(c)**     **(d)**

T: News Text    I: News Image    C: News Comments    A: News Audio    V: News Video

D: Dynamic Feature     S: Static Feature     Z: Multimodal Feature     Y: Predicted Label

**Figure 2: Four model's learning and reasoning patterns of multimodal short videos. (a) Model's expected learning and reasoning pattern of short video(b) Dynamic bias mitigation of model's learning and reasoning pattern; (c) Static bias mitigation of model's learning and reasoning pattern; (c) Social bias mitigation of model's learning and reasoning pattern.**

Existing detection models are expected to learn multimodal information and reason its label correlation as a causal graph, as depicted in Figure 2(a). However, these models are plagued by multiple biases, which we summarize as static bias, dynamic bias and social bias. These biases lead to spurious label correlations, as illustrated in Figures 2(b), 2(c) and 2(d). **1) Static Bias: T, I $\rightarrow$ S $\rightarrow$ Y** branch. Static content bias occurs when a model learns excessive correlation between key textual, visual contents and labels [2, 21]. From real-world observations, we notice that people often publish news or tweets using the same or similar textual and visual contents, such as those containing "Trump". These contents in news articles or tweets often carry contradictory verification labels. For instance, the mapping of "Donald Trump" to the "fake category" is much more prevalent compared to when he was the president [38]. This phenomenon highlights the direct spurious causal correlation of labels with the textual and visual contents, indicating a form of static bias in detecting fake news [2]. **2) Dynamic Bias**: **V, A $\rightarrow$ D $\rightarrow$ Y** branch. Sneaky fake news creators often manipulate news videos by incorporating old clips of dramatic conflicts or exercises, altering the audio to deceive the public and create the illusion of authenticity, particularly in the context of the Ukraine war [26]. Such news videos have often been mistaken for real in the past. However, these instances can lead models to learn direct spurious causal correlations between the video-audio content and the associated labels. **3) Social Bias: C $\rightarrow$ Y** branch. The study [33] demonstrates that models usually suffer from noisy contextual information, such as comments. Consequently, existing models may unfairly reason with biased context and make incorrect predictions. As illustrated in Figure 2(d), news comments may introduce spurious causal correlations in detecting fake news. This can be interpreted as models opting to bypass the application of the multimodal feature $Z$ and instead relying solely on news comments as a shortcut to verify news authenticity.

To address these issues, we propose a Multi-View Multimodal Debiasing (MMVD) Framework, which makes the first attempt to mitigate various multimodal biases for fake news video detection. In particular, inspired by people's misleading situations by multimodal short videos, we summarize three cognitive biases: static, dynamic

and social biases. Specifically, the MMVD mitigates static biases by exploiting a counterfactual scenario for estimating the direct influences. Further, MMVD designs a coherence constrain reasoning strategy to probe the dynamic bias caused by video-audio features through constraint strategy. Different from previous studies, the MMVD applies a causal reasoning learning strategy by causally removing the direct biases of the social comments for prediction. Therefore, our MMVD can exploit a multi-view causal reasoning strategy to learn unbiased dependencies within the cognitive biases, thus enhancing the unbiased prediction of multimodal videos. The extensive experimental results show that the MMVD could improve the detection performance of multimodal fake news video detection. Studies also confirm that our MMVD can mitigate multiple biases on complex real-world scenarios and improve generalization ability of multimodal models. The contributions of MMVD are summarized as follows:

- A new paradigm of Multimodal Multi-View Debiasing(MMVD) framework is proposed, which makes the first attempt to mitigate various multimodal biases for fake news video detection.
- In our MMVD, the coherence constrain reasoning strategy can not only infer the coherence between video and audio, but also evaluate and mitigate the dynamic bias between video and audio of news.
- The extensive experimental results on two real-world datasets show that the MMVD could improve the performance of fake news video detection. Studies also confirm that our MMVD can mitigate multiple biases on complex real-world scenarios and improve generalization ability of fake news video detection.

## 2 RELATED WORK

### 2.1 Fake News Video Detection

Fake news video detection is a newly emerged research area that has garnered the attention of many researchers. Numerous outstanding methods have been developed in recent years, with several traditional approaches being applied to the task of detecting fake news

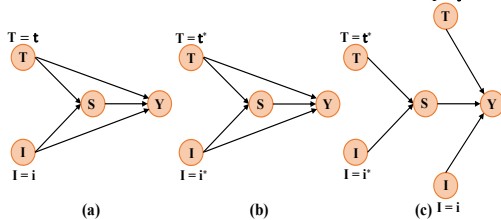

**Figure 3: Example of causal graph where T, I, S and Y denote textual cause, visual cause, mediating variable and effect, respectively. ∗ denotes the reference values.**

videos. Olga et al.[17] presented the first fake news video dataset, FVC-2018, using a support vector machine for metadata and comment descriptor vectors to detect fake news [18]. Serrano et al. [22] extracted contextual features from video titles and comments and selected three machine learning models, such as, Logistic Regression(LR), Support Vector Machine(SVM) and Random Forest(RF) as classification models. Hou et al.[9] introduced extra information, such as, audio and audience feedback and speech signals to provide comprehensive dimension to distinguish fake news.

Compared to these machine learning methods, deep learning methods are superior in terms of their strong ability to extract features of news. Li et al. [11] integrated content, uploader and environment features to construct a Convolutional Neural Network for fake news video detection. Palod et al. [16] combined UCNet and LSTM networks for modeling comments to detect fake news. FANVN [3] identified differences in stance through differences in topic distribution between titles/descriptions and comments and constructed an adversarial neural network to efficiently extract topic agnostic features. TikTec exploited the captions to accurately capture the key information from the distractive video content and effectively learns the composed misinformation that is jointly conveyed by the visual and audio content [23]. TwtrDetective incorporated cross-media consistency to identify fake news video[13]. Further, Qi et al. [19] constructed a largest Chinese dataset of short fake news videos and proposed a new multimodal detection model named SV-FEND, which extracts multimodal features and integrates the multimodal features using Transformer. NEED [20] applied the GAT network to integrate extra event and debunking video information for comprehensive fake news video detection.

## 2.2 Causal Inference

Causal inference contains counterfactual reasoning and causal reasoning strategies, which is a crucial causal approach used in natural language processing [12, 36] and social network analysis [2]. Causal inference is an effective way to remove the confounding causality in the prediction process. As for fake news detection task, Zhu et al [38] removed the textual entity bias by jointly considering the direct causal relation of the entity and the content on prediction. DCE-RD mitigated the propagation bias by exploiting multi-view counterfactual evidence in an event graph [37]. Wu et al [34] made unbiased fake news detection by jointly conduct conventional predictions and counterfactual predictions based on the intervened evidence. For complex video-based task, Lv et al. [14] applied a

counterfactual cross-modality reasoning method to discover and mitigate the spurious correlation in video moment localization. CLUE model [27] mitigates the bias in video-based sentiment analysis and strengthens the generalization ability of the models on the OOD datasets.

Existing fake news video detection models focused on integrating multimodal information of short videos and predict their false correlation labels. Further, they turned to introduce additional information, such as event and debunking video information, ignoring to exploit the nature of the misclassification. In our study, we make the first attempt to enhance detection and generalization performance by mitigating the static, dynamic and social biases when models are biased by historical and misleading multimodal information.

## 3 PROBLEM DEFINITION

In this study, we approach the fake news video detection task as a binary classification problem. Our focus is on designing a model, denoted as $Y(\cdot)$, aimed at mitigating static, dynamic and social biases within a causal graph framework to achieve unbiased fake news video detection.

### 3.1 Multimodal Causal Graph

Let $\mathcal{G} = \{\mathcal{V}, \mathcal{E}\}$ be a multimodal causal graph, where $\mathcal{V} = \{T, I, C, A, V, S, D, Z, Y\}$ contains nine-type nodes; $\mathcal{E}$ denotes four cause-effect paths: (1) Audio, Video, Text, Image, Comment-to-Multimodal Feature-to-Label ($A, V, T, I, C \to Z \to Y$); (2) Text, Image-to-Static Feature-to-Label ($T, I \to S \to Y$); (3) Audio, Video-to-Dynamic Feature-to-Label ($A, V \to D \to Y$); (4) Comments-to-Label ($C \to Y$). As shown in Figure 2, fake news video detection models suffer from dynamic($D$), static($S$) and social($C$) biases, which results in degrading the model's detection and generalization ability. In this study, we aim to mitigate these biases for unbiased detection.

### 3.2 Multimodal Counterfactual Reasoning

Counterfactual reasoning, an important causal inference method, can give a model the ability to imagine the counterfactual contents, leading to make an unbiased prediction. Inspired Counterfactual reasoning theory, we derive the calculation procedure for the multimodal counterfactual reasoning. For instance, Figure 3 represents the situation how multimodal counterfactual reasoning estimates and removes the direct effect of $T, I$ on $Y$. As shown in Figure 3(a), in traditional studies, the prediction is directly affected by textual and visual contents simultaneously in the causal graph $\mathcal{G}$. Figure 3(a) is the factual situation and the prediction of $Y$ can be expressed as $Y = Y(T = t, I = i, S = S(t, i))$. Figure 3(b) is a counterfactual situation via reversing the textual and visual contents, expressed as $Y^* = Y(T = t^*, I = i^*, S = S(t^*, i^*))$. Illustrated in Figure 3(a) and 3(b), we define the total effect (TE) of $T = t, I = i$ on $Y$ as:

$$\begin{aligned} TE = \ &Y(T = t, I = i, S = S(t, i)) \\ &- Y(T = t^*, I = i^*, S = S(t^*, i^*)). \end{aligned} \tag{1}$$

Here, $TE$ can be viewed as the comparison between two mutually contrary results of $T, I$, such as $T = t, I = i$ and $T = t^*, I = i^*$. The total effect (TE) consists of the natural direct effect (NDE) and the total indirect effect (TIE), which can be denoted as $TE = NDE + TIE$. As shown in Figure 3(c), while the mediating variable $S$ is blocked,

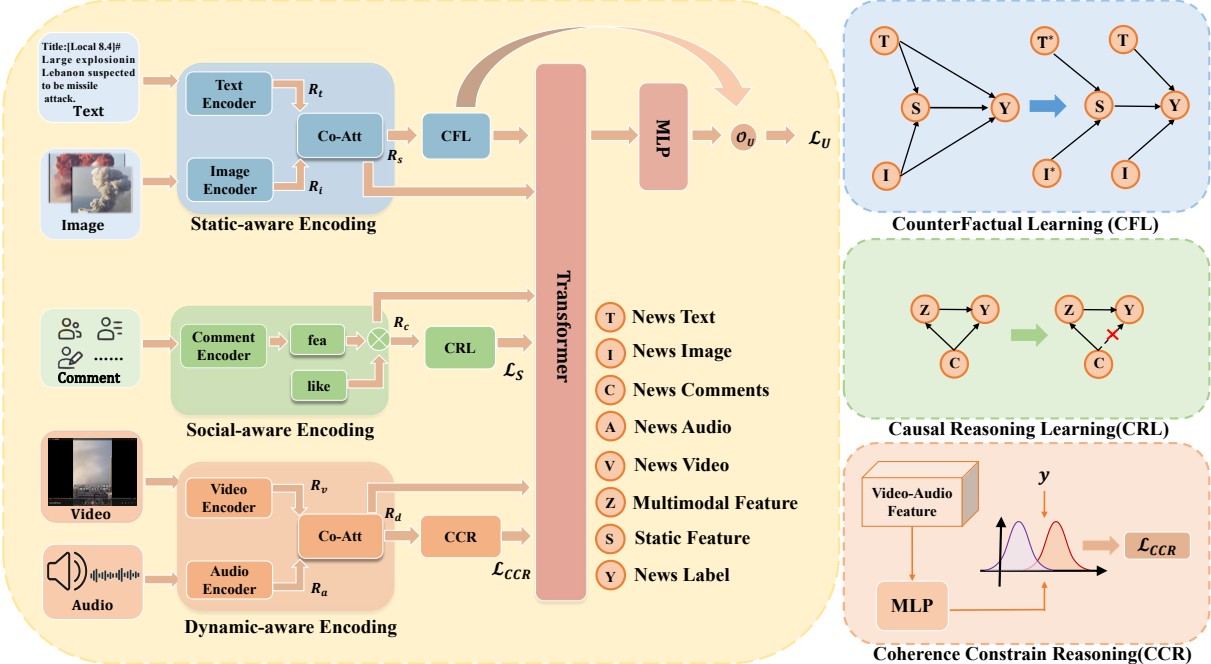

**Figure 4: Overview of proposed Multimodal Multi-View Debiasing framework. The CFL, CCR and CRL mitigate the static, dynamic and social biases during multimodal fusion, respectively. Then the MMVD is learned to determine whether the news video is fake or not.**

$NDE$ represents the natural direct effect of $T = t, I = i$ on $Y$,

$$NDE = Y(T = t, I = i, S = S(t^*, i^*))$$
$$- Y(T = t^*, I = i^*, S = S(t^*, i^*)), \quad (2)$$

where $Y(T = t, I = i, S = S(t^*, i^*))$ is the calculation result under counterfactual situation. Therefore, the total indirect effect (TIE) is considered as the debiased result, which can be expressed as:

$$TIE = TE - NDE,$$
$$= Y(T = t, I = i, S = S(t, i)) \quad (3)$$
$$- Y(T = t, I = i, S = S(t^*, i^*)).$$

In this study, we leverage the *TIE* to mitigate textual and visual biases for unbiased fake news video detection.

## 4 METHODOLOGY

### 4.1 Framework Overview

The MMVD aims to mitigate global and local biases of multimodal news for fake news video detection. The overall framework is illustrated in Figure 4, which consists of four main procedures: the *Counterfactual Reasoning Learning Strategy*, *Coherent Constrain Reasoning Strategy*, *Causal Reasoning Learning Strategy*, *Training Stage*. During multimodal fusion, there are three branches corresponding to the biases mitigation from three views, including static, dynamic and social biases mitigation. Finally, the MMVD jointly mitigates static, dynamic and social biases of multimodal news for fake news video detection.

### 4.2 Inputs Encoding

#### 4.2.1 Static-aware Encoding. **Text:** For this task, news titles and video captions play an important role in fake news video detection. We create synthesized text by concatenating them, which is then fed into the pre-training BERT [6] to extract textual feature, denoted as $R_t$.

**Image:** Keyframes in news videos often offer viewers their initial impressions and encapsulate vital static visual information. To analyze this content, we use the pre-trained VGG19 [25] model to extract the image feature $R_i$.

Static-aware information of news videos convey the first impression to readers where exaggerated historical titles and images can easily attract attentions and deceive readers. To integrate the static-aware information, we use the cross-attention to integrate textual and visual information between $R_t$ and $R_i$. The calculation process of static feature $R_s$ can be denoted as:

$$R_s = att(R_t, R_i). \quad (4)$$

#### 4.2.2 Dynamic-aware Encoding. **Video:** Video has unique spatio-temporal characteristics [35], showing communication effect, emotional expression ability, which are higher than other modalities. Therefore, video clips are more capable of giving an deep impression of the video [4]. To analyze motion trajectories within videos, we use the pre-trained C3D [30] for feature extraction and apply averaging operation to aggregate the video features $R_v$.

**Audio:** Audio contains speech and background music, which appears simultaneously with video to deliver comprehensive messages. The textual component of audio enhances the semantic understanding of video content [11], while variations in volume and tone offer insights into the speaker's emotions and intentions [19]. We isolate the audio track from videos and feed it into a pre-trained VGGish [7] model to extract audio feature $R_a$.

Dynamic-aware information of news videos conveys the deep impression to readers and exaggerated videos and audios can easily attract and deceive readers. To integrate the dynamic-aware information, we use the cross-attention to integrate video and audio information between $R_v$ and $R_a$. The calculation process of dynamic feature $R_d$ can be expressed as:

$$R_d = att(R_v, R_a). \qquad (5)$$

*4.2.3 Social-aware Encoding.* **Comment:** Social comments reflect the opinions or sentiments of social media users and play a significant role in predicting the authenticity of news videos [23]. To identify the most representative comments, we utilized the number of likes as a proxy for their relevance or popularity, extracting the top-k comments based on this criterion for further analysis. Given that comments are primarily textual, we employed a pre-trained BERT model [6] to extract their features, denoted as $R_c$.

## 4.3 Coherent Constrain Reasoning

Sneaky fake news creators often create a news video with an old clip of a dramatic conflict or exercise and edit the audio for cheating public to make it look as real as possible, especially in the Ukraine war [26]. These original and historcal news videos may often be learned as real by existing models, which may make model learn direct spurious causal correlation between video-audio features and labels. To address this issue, we propose Coherent Constrain Reasoning(CCR) to maximise the discrimination between video-audio features and labels to mitigate this bias.

Inspired by information bottleneck [29], CCR is designed to refine the intermediate feature $z$ to be maximally discriminative of the target label while minimizing its retention of unnecessary input information $x = R_d$. Ideally, $z$ becomes concise yet sufficiently discriminative for predicting the target label. Let $y \in \{0, 1\}$ represent the classification label, where $y = 0$ indicates the video is fake news, while $y = 1$ denotes real news. CCR's objective is to maximize the mutual information between $z$ and the label $y$, while minimizing the mutual information between $z$ and the input $x$. The CCR objective function is formally defined as:

$$F_{CCR} = I(y; z) - \beta \cdot I(x; z). \qquad (6)$$

Based on deep variational information bottleneck theory [1], we aim to optimize the lower bound $L_{CCR}$ of $F_{CCR}$, thereby maximizing $F_{CCR}$ as follows:

$$\begin{aligned} F_{CCR} = I(y; z) - \beta \cdot I(x; z) &\geq L_{CCR} \\ &= \mathbb{E}_{(x,y) \sim p(x,y), z \sim p(z|x)} [\log q(y|z) \\ &\quad - \beta \cdot KL(p(z|x)||q(x)]. \end{aligned} \qquad (7)$$

To implement this optimization, we utilize a MultiLayer Perceptron (MLP) [15] to model $p(z|x)$ as a Gaussian distribution, with

the distribution's mean and variance computed by the MLP:

$$\begin{aligned} p(z|x) &= \mathcal{N}(\mu(f; \theta_\mu), \Sigma(f; \theta_\Sigma)) \\ &= \mathcal{N}(\mu_z, \Sigma_z), \end{aligned} \qquad (8)$$

where $\mu$ and $\Sigma$, parameterized by $\theta_\mu$ and $\theta_\Sigma$, respectively, denote the mean and variance of $p(z|x)$. However, the addition of MLP leads to a randomness in the computation of the parameter gradient, making the updating of the parameters difficult. To solve this issue, we refer to the reparameterization trick [32] to get $z$:

$$z = \mu_z + \Sigma_z \times \varepsilon, \qquad (9)$$

where $\varepsilon \sim \mathcal{N}(0, I_v)$ is a standard normal Gaussian distribution and $I_v$ is the identity vector. The difficulty of parameter updating can be greatly reduced by using the reparameterization trick.

We follow the previous work [15] and assume that each element in the vector $z$ is independent of each other. Moreover, for classification task, $q(y|z)$ can be formulated as:

$$\log q(y|z) = y \log(\sigma(MLP(z; \theta_d))) = y \log \hat{y}, \qquad (10)$$

where $MLP$ is a decoder parameterized by $\theta_d$ and $\hat{y}$ is the prediction result. In addition, the approximated marginal distribution of the intermediate feature $z$ is often assumed to be a standard normal Gaussian distribution, $q(z) \sim \mathcal{N}(0, I_v)$ [15]. Thus, $KL(p(z|x)||q(z))$ can be calculated as:

$$KL(p(z|x)||q(z)) = KL(\mathcal{N}(\mu_z, \sigma_z)||\mathcal{N}(0, I_v)). \qquad (11)$$

Note that we assume the reparameterization of $p(z|x)$ and $q(z)$ allows for computation of an analytic KL-divergence.

Finally, the integral over $x$, $z$ and $y$ can be approximated by Monte Carlo sampling [24], leading to the final loss function for CCR $\mathcal{L}_{CCR}$ can be expressed as:

$$\mathcal{L}_{CCR} = \sum_{y \in Y^l} y \log \hat{y} - \beta \cdot KL(\mathcal{N}(\mu_z, \Sigma_z)||\mathcal{N}(0, I_v)), \qquad (12)$$

where, $Y^l$ is the set of ground truth labels.

## 4.4 Counterfactual Reasoning Learning

Static bias occurs when a model learns excessive correlation between key textual, visual contents and labels [2, 21]. Counterfactual reasoning learning strategy can help model to imagine a counterfactual static world to estimate the direct effects of static information in fake news video detection.

Specially, the textual content words contains strong emotional words, keywords, such as "black" or "disappointed", leading to spurious correlation between textual contents and labels [12]. Meanwhile, real-world models still suffer from visual bias, where the news with convincing or extreme images are biased to be predicted as real or fake [2]. To mitigate this static bias, we construct textual and visual reference situations by blocking textual and visual inputs $T^*, I^*$ to estimate direct influence of $T, I$ on $Y$. Therefore, we can gain the natural direct effect (NDE) $O_{NDE}$ of $T, I$ on $Y$:

$$\begin{aligned} O_{NDE} = &f_s(T, I, S = S(T^*, I^*)) \\ &- f_s(T^*, I^*, S = S(T^*, I^*)). \end{aligned} \qquad (13)$$

Here, $f_s$ is a text-image prediction model, and $S(T^*, I^*)$ denotes static feature of $T^*, I^*$. Generally, we discover the global logit prediction $O_C$ on fusing $R_s, R_d, R_c$ by Transformer:

$$O_C = Transformer(R_s, R_d, R_c). \qquad (14)$$

Base on multimodal counterfactual reasoning theory, the calculation of the total indirect effect(TIE) by causally considering $O_C$ and $O_{NDE}$. The process can be represented as:

$$O_R = O_C - O_{NDE}, \qquad (15)$$

where $O_R$ denotes the reasoning prediction, which is unbiased text-image prediction.

## 4.5 Causal Reasoning Learning

The existing models often suffered biases caused by the responses [33] from social platforms. The influence always comes from noisy and misleading responses that bring about spurious relationships, which results in learning social biased information. To remove the social biased influence, we design a causal reasoning learning strategy to mitigate biases from external information on social platforms.

To explicitly model the possible social-biased influence, we train an social-biased model only with comment feature $R_c$ as input and exploit social prediction $O_C$:

$$O_C = f_C(R_c), \qquad (16)$$

where $f_C$ is a fully-connected layer.

We then use unbiased prediction $O_U$ to causally consider static, dynamic and social biases by linking with the two parts $O_R$ and $O_S$, which can be expressed as:

$$O_U = \sigma(\gamma(O_R) + (1 - \gamma)O_S), \qquad (17)$$

where $\sigma(\cdot)$ indicates the Sigmoid function and $\gamma$ is a hyper-parameter to balance the $O_R$ and $O_S$. Then, we apply the unbiased prediction $O_U$ with the cross-entropy loss:

$$\mathcal{L}_U = \sum_{y \in Y^l} -y\log(O_U) - (1 - y)\log(1 - O_U), \qquad (18)$$

where $Y^l$ is the set of ground truth labels. To achieve the mitigating effect of social comments, we use an auxiliary loss, which focuses on the bias of comments:

$$\mathcal{L}_S = \sum_{y \in Y^l} -y\log(\sigma(O_S)) - (1 - y)\log(1 - \sigma(O_S)). \qquad (19)$$

## 4.6 Training Stage

Training the above these strategies jointly, the final loss function $\mathcal{L}_f$ can be expressed as:

$$\mathcal{L}_f = \mathcal{L}_U + \lambda_1 \mathcal{L}_S - \lambda_2 \mathcal{L}_{CCR} \qquad (20)$$

where the $\lambda_1, \lambda_2$ are hyper-parameters and we minimize the total training loss $\mathcal{L}_f$ to fine-tune all parameters. This training procedure can enable the models to learn less biased multimodal information for fake news video detection.

# 5 EXPERIMENTS

## 5.1 Datasets and Evaluation Metrics

*5.1.1 Datasets.* To validate the superiority of MMVD, we experiment on two competitive datasets FakeSV and FVC. The details of datasets are described as follows:

- **FakeSV [19]**: FakeSV is the largest Chinese fake news short video dataset, which contains multimodal information, such as news video, audio, text, image, user comments and publisher profiles. These short videos were verified by Chinese official fact-checking sites between January 2019 and January 2022.
- **FVC [17]**: The FVC dataset, which was developed in the In VIDeo Veritas(InVID) project, is composed of fake and real news videos of various topics, including politics, sports and accidents. This study mainly focuses on short videos on YouTube.

*5.1.2 Evaluation Task Settings and Metrics.* To ensure fairness of experiments, our evaluations are conducted by applying five-fold. For each fold, the dataset is split as training and testing sets with a ratio of 4:1 on FakeSV and FVC datasets as [17, 19]. We use maximum length of the news text as 100 and the Bert-based uncased [6] for the datasets of FVC and Chinese pre-trained BERT with Whole Word Masking [5] for FakeSV dataset, respectively. In the MMVD, we set $\lambda_1 = 0.1, \lambda_2 = 0.1$ and use Adam as the optimizer, learning rate as 5e-5, batch_size as 128. According to the metrics accuracy(ACC), macro $F_1$, macro Precision(Pre) and macro Recall(Rec), the detection performances of baselines and our MMVD are shown in Table 1. Their largest values are emphasized in bold. The experiments were conducted on NVIDIA 3090Ti GPUs.

## 5.2 Performance Results

In this study, we conduct comparison experiments to verify the performance of the proposed MMVD. According to the metrics Accuracy, $F_1$, Precision and Recall, the detection performances of MMVD and comparison models on the datasets of FakeSV and FVC are shown in Table 1. Among the nine detection models, the MGCL achieves the best detection performance on two datasets. Specifically, our MMVD outperforms the corresponding state-of-the-art methods, SV-FEND by at least 3.33% and 4.57% in Accuracy value on FakeSV and FVC datasets, respectively. Fake news video detection methods that leverage video information, such as TikTec [23] and SV-FEND [19], generally outperform the models that don't. This again reaffirms the importance of analyzing the multimodal information for robust fake news video detection.

The multimodal methods, such as (Hou et al.) [10] and (Serrano et al.) [22], outperform the unimodal methods, VGGish, VGG-19 and C3D. This reaffirms the significance of leveraging the multiple interactions of different modalities for fake news video detection. With an increasingly complexity of social network, several methods introduced the comments from real world (FANVN [3]), showing more comprehensive modeling ability for news video verification. Our MMVD models audio, video, textual, visual contents and social comments into a unified framework, which offers us the possibility to make the most of multiple structures of news on social media and improve the detection performance.

**Table 1: Experimental results of baselines and the proposed MMVD on FakeSV and FVC datasets.**

| | FakeSV | | | | FVC | | | |
|---|---|---|---|---|---|---|---|---|
| Method | Acc | $F_1$ | Pre | Rec | Acc | $F_1$ | Pre | Rec |
| VGGish+SVM(Audio) | 61.25 | 61.31 | 61.24 | 61.33 | 58.44 | 58.61 | 58.48 | 58.63 |
| Bert+Att(Comment) | 62.74 | 62.75 | 62.71 | 62.76 | 61.70 | 61.76 | 61.81 | 61.72 |
| VGG19+Att(Image) | 68.53 | 68.51 | 68.53 | 68.50 | 65.79 | 65.81 | 65.49 | 66.08 |
| C3D+Att(Video) | 70.26 | 70.24 | 70.25 | 70.25 | 71.81 | 71.72 | 71.89 | 71.85 |
| Bert+Att(Text) | 74.31 | 74.35 | 74.30 | 74.39 | 76.37 | 76.35 | 76.39 | 76.33 |
| (Hou et al.)[10] | 68. 64 | 68. 01 | 70. 24 | 68. 63 | 66. 87 | 66. 74 | 67. 15 | 66. 34 |
| (Serrano et al.)[22] | 71. 45 | 71. 45 | 71. 47 | 71. 45 | 71. 06 | 72. 00 | 72. 64 | 71. 38 |
| TikTec [23] | 75. 07 | 75. 04 | 75. 18 | 75. 07 | 77. 02 | 73. 95 | 74. 24 | 73. 67 |
| FANVN [3] | 75. 04 | 75. 02 | 75. 11 | 75. 04 | 85. 81 | 85. 32 | 85. 20 | 85. 44 |
| SV-FEND [19] | 79. 31 | 79. 24 | 79. 62 | 79. 31 | 84. 71 | 85. 37 | 84. 25 | 86. 53 |
| **MMVD** | **82. 64** | **82. 63** | **82. 63** | **82. 73** | **89. 28** | **90. 36** | **90. 27** | **90. 46** |

Both SV-FEND[19] and MMVD employ audio-video, text-image contents and social comments for fake news video detection. MMVD outperforms SV-FEND on all two datasets, providing empirical evidence that our multimodal multi-view debiasing framework can mitigate the dynamic, static and social biases for multimodal fusion, which is more robust for fake news video detection.

## 5.3 Ablation Study

We design four ablation experiments to evaluate the effectiveness of components in MMVD. Specifically, we design several internal models with inputs and strategies removed. The performance comparisons are shown in Table 2.

- **w/o Text**: We remove the news text of inputs.
- **w/o Vis**: We remove the news image of inputs.
- **w/o Video**: We remove the news video of inputs.
- **w/o Audio**: We remove the news audio of inputs.
- **w/o Comment**: We remove the news comments of inputs.
- **w/o CFL**: We remove the counterfactual reasoning strategy for fake news video detection.
- **w/o CRL**: We remove the causal reasoning learning strategy for fake news video detection.
- **w/o CCR**: We remove the coherence constrain reasoning strategy for fake news video detection.

**Importance of Leveraging Multimodal Information:** To evaluate the effectiveness of leveraging multimodal information, we design five internal models by removing the news text, image, video, audio, comments of inputs. The results in Table 2 reveals that removing any inputs of multimodal news would lead to performance drop, while the news text structure is most critical to model performance, resulting in an 7.38% drop in model Accuracy on FakeSV dataset. As a result, the MMVD completely apply the multimodal information for multimodal representation learning for enhancing fake news video detection.

**Significance of Mitigating the Various Biases:** To investigate the Significance of mitigating the static, dynamic and social biases, we remove the CCR, CFR and CRL strategies, respectively. Results in Table 2 demonstrate that the MMVD outperforms all internal models, validating the effectiveness of the CCR, CFR and CRL strategies that enables exploiting causal correlations of multimodal

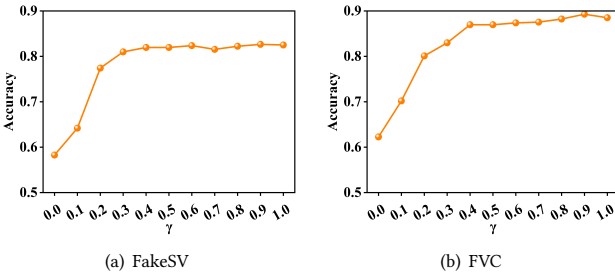

(a) FakeSV       (b) FVC

**Figure 5: Performance of sensitive analysis on two datasets.**

information and prediction results. This is also verified that our multimodal multi-view debiasing strategy can guide our MMVD to learn less biased multimodal information for improving detection performance.

## 5.4 Sensitive Analysis

Different from existing methods, our MMVD found that the detection of news videos is effected by dynamic, static and social biases of multimodal news video. Thus, the hyper-parameters $\gamma$ plays the most crucial role to balance the prediction between news contents and social comments. The $\gamma$ value represents the importance of reasoning prediction in the detection process. On the datasets of FakeSV and FVC, along with the change in the $\gamma$ value, the changes in evaluation performances are shown in Figure 5.

As is shown in Figure 5, we observe that the general evaluation performance of the MMVD is strongly correlated with $\gamma$ in most cases, which further suggests that causally considering reasoning prediction and social prediction in detecting news video is beneficial. A larger $\gamma$ value indicates that the reasoning prediction plays a more important role in detecting fake news video. When $\gamma = 0.9$ on FakeSV and FVC datasets, our MMVD achieved the best performance. The cooperation between the prediction of reasoning and social predictions in our MMVD has the potential to benefit the detection process. In actual application, we can select the optimal $\gamma$ value to promote the detection performance.

**Table 2: Experimental results of baselines and the proposed MMVD on FakeSV and FVC datasets.**

| Method | FakeSV | | | | FVC | | | |
|---|---|---|---|---|---|---|---|---|
| | Acc | $F_1$ | Pre | Rec | Acc | $F_1$ | Pre | Rec |
| w/o Audio | 78.41 | 78.39 | 78.33 | 78.47 | 78.87 | 78.81 | 78.83 | 78.80 |
| w/o Video | 78.17 | 78.94 | 78.21 | 78.97 | 79.43 | 79.41 | 79.42 | 79.40 |
| w/o Image | 77.97 | 77.84 | 77.89 | 77.92 | 74.56 | 74.40 | 75.58 | 74.49 |
| w/o Text | 75.26 | 75.67 | 75.14 | 75.31 | 72.19 | 72.34 | 72.18 | 72.46 |
| w/o Comment | 78.33 | 78.37 | 78.43 | 78.31 | 80.87 | 80.86 | 80.87 | 80.84 |
| w/o CFL | 80.99 | 80.97 | 80.97 | 81.11 | 88.74 | 88.75 | 88.75 | 88.73 |
| w/o CRL | 81.96 | 81.93 | 81.94 | 82.07 | 88.26 | 88.24 | 88.15 | 88.36 |
| w/o CCR | 81.72 | 81.80 | 81.79 | 81.83 | 87.18 | 87.15 | 87.15 | 87.14 |
| **MMVD** | **82.64** | **82.63** | **82.63** | **82.73** | **89.28** | **90.36** | **90.27** | **90.46** |

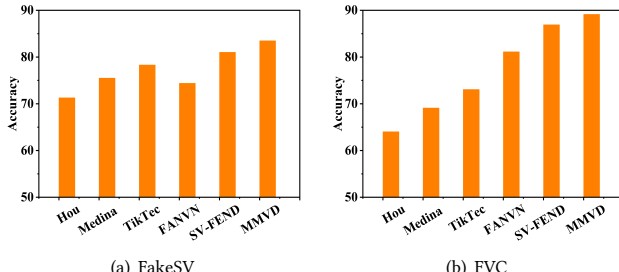

(a) FakeSV (b) FVC

**Figure 6: Performance comparison of temporal analysis on two datasets.**

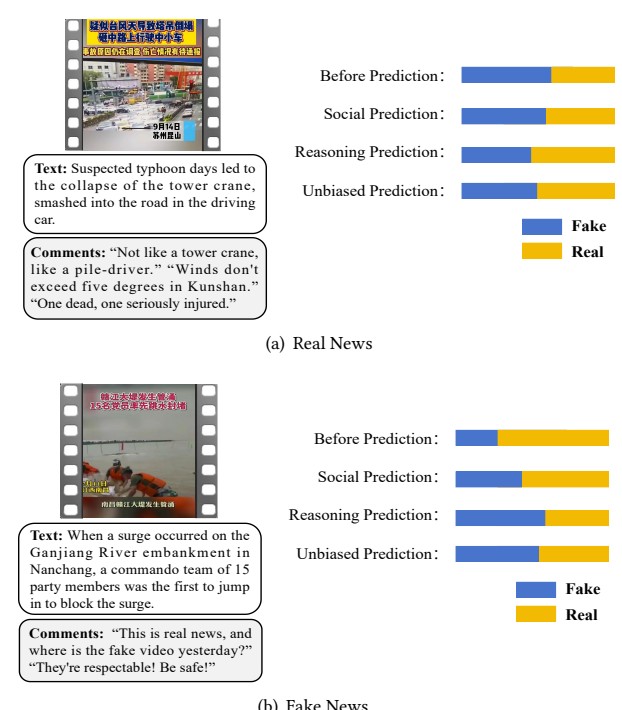

**Figure 7: Case Study on FakeSV dataset.**

## 5.5 Temporal Analysis

The models creates an biased impression because they learn the relationship between historical short videos and labels, leading to bias in detecting fake news in the future. To verify our MMVD's bias mitigation ability, we temporally partitioned the datasets in the ratio of 70%: 15%: 15% in the order of video publication time as training, validation and testing sets, respectively, to evaluate the model's bias mitigation ability to detect fake news videos in temporal order.

As shown in Figure 6, MMVD still achieved the best performance in detecting fake news videos in temporal order. This phenomenon can be expressed that our MMVD, applying the CCR, CFL and CFL, can mitigate the biases of fake news videos, which come from historical and misleading multimodal information.

## 5.6 Case Study

We provide a case study that visualizes the results of before (Before Prediction), intermediate (Reasoning Prediction and Social Prediction) and after (Unbiased Prediction) using our MMVD and provides an explanation for the strong performance of MMVD. We randomly selected two news samples from FakeSV dataset and focused on prediction changes, depicted in Figure 7. As shown in Figure 7, based on reasoning and social predictions by leveraging counterfactual reasoning learning and causal reasoning learning strategies, the MMVD corrects the prediction results. Therefore, the MMVD framework enables the model to focus on learning less biased multimodal information and provide more reliable results. This studies also confirm that MMVD can enhance the generalization ability of

fake news video detection by alleviating the dynamic, static and social biases on complex real-world scenarios.

## 6 CONCLUSION

In this study, we recognize the evaluation issue of biased fake news video detection. To solve this issue, we propose a Multimodal Multi-View Debiasing(MMVD) framework, which makes the first attempt to mitigate various multimodal biases of multimodal samples for fake news video detection. Inspired by people's misleading situations by multimodal short videos, we summarize three cognitive biases: static, dynamic and social biases. MMVD designs a multi-view causal reasoning strategy to learn unbiased dependencies within the cognitive biases. The extensive experimental results show that the MMVD could improve the detection performance and generalization ability.

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
