# OpenReview forum: "Mitigating World Biases: A Multimodal Multi-View Debiasing Framework for Fake News Video Detection"
_acmmm.org/ACMMM/2024/Conference — MM2024 Poster_

### Official Review · Reviewer_6mLH · 2024-05-14

**Rating:** 4
**Confidence:** 3

**Summary:**

This paper focuses on the issue of bias in fake news video detection and proposes a Multimodal MultiView Debiasing (MMVD) framework. The authors identify three types of cognitive biases: static, dynamic, and social. Counterfactual Reasoning Learning Strategy, Coherent Constraint Reasoning Strategy, and Causal Reasoning Learning Strategy are applied, respectively, to mitigate these biases in fake news video detection.

**Strengths:**

1. This article focuses on the detection of fake news in short videos, which is a practical problem of significant importance in the real world.
2. The paper is well-structured and adheres to academic writing standards.
3. The authors provide their code, ensuring the reproducibility of their methods.

**Limitations:**

1. The correlation between some technical approaches and the biases they are intended to address is not very strong. For instance, employing information bottleneck theory to alleviate dynamic bias seems puzzling without a clear causal explanation. The three strategies used in MMVD are not novel designs and have been employed in related fields [1, 2, 3].  A more pointed explanation of why these particular strategies are effective in video fake news detection would help enhance the paper.
2. Some methodological descriptions lack clarity, such as how terms like T* and I* are derived.
3. The case studies provided are insufficient. Direct examples could better demonstrate the effectiveness of MMVD. Currently, the cases only show MMVD’s capability of mitigating social bias. The authors could enhance their manuscript by including additional examples demonstrating how MMVD improves detection performance by alleviating static or dynamic biases.

[1] Zhu, Yongchun, et al. "Generalizing to the future: Mitigating entity bias in fake news detection." Proceedings of the 45th International ACM SIGIR Conference on Research and Development in Information Retrieval. 2022.
[2] Hu, Linmei, et al. "Causal inference for leveraging image-text matching bias in multi-modal fake news detection." IEEE Transactions on Knowledge and Data Engineering 35.11 (2022): 11141-11152.
[3] Mai, Sijie, Ying Zeng, and Haifeng Hu. "Multimodal information bottleneck: Learning minimal sufficient unimodal and multimodal representations." IEEE Transactions on Multimedia (2022).

**Suitability:**

3

---

### Official Review · Reviewer_e1P9 · 2024-05-20

**Rating:** 3
**Confidence:** 2

**Summary:**

Short videos have become a prominent platform for sharing information, but they also harbor fake news. Current detection models often rely on superficial correlations between different modalities, leading to degraded detection and generalization capabilities. To address this, they propose the Multimodal Multi-View Debiasing (MMVD) framework, which aims to mitigate biases inherent in multimodal data. MMVD leverages a multi-view causal reasoning strategy to understand unbiased dependencies within cognitive biases, thus improving the detection accuracy of fake news videos. Experimental results demonstrate MMVD's effectiveness in enhancing detection performance and mitigating biases in real-world scenarios, thereby improving generalization.

**Strengths:**

1. Experimental results show that the proposed method is better than the competing methods by health margin.
2. The ablation study seems sufficient and shows the effectiveness of the designed modules.
3. Innovative combination of causal reasoning with fake news video detection。

**Limitations:**

1. It is not clear how the "World Biases" described in the paper specifically affect model performance and how it helps.

2. The improvement of MMVD in Table 2 seems convincing, but it might be due to more parameters and complexity of the model.

3. Can you explain in detail how causal reasoning can help with multimodal learning of fake news? such as core benefits over traditional methods

4. Comparisons can be made with more advanced methods such as [1].

5. How to understand the "generalization capabilities" of MMVD: such as zero-shot?

[1] Qi P, Zhao Y, Shen Y, et al. Two heads are better than one: Improving fake news video detection by correlating with neighbors[J]. arXiv preprint arXiv:2306.05241, 2023.

**Suitability:**

2

---

### Official Review · Reviewer_NaAx · 2024-05-25

**Rating:** 3
**Confidence:** 3

**Summary:**

This paper proposes a multimodal mlti-View debiasing framework to mitigate three cognitive biases (static, dynamic and social) for fake
news video detection.

**Strengths:**

The static, dynamic and social biases are formulated during multimodal fusion.

**Limitations:**

1) The description of Social-aware Encoding is not clear.
2) Most of the comparison method are not latest, e.g., CIKM 2021, ICMI 2019, ACL workshop 2020, IEEE conference on big data 2021.

**Suitability:**

3

---

### Official Review · Reviewer_c3tc · 2024-06-04

**Rating:** 5
**Confidence:** 3

**Summary:**

This work focuses on the problem of biases in fake news video detection and proposes a novel framework called Multimodal Multi-View Debiasing (MMVD) to address this issue. The MMVD framework aims to mitigate global and local biases in multimodal news for fake news video detection by incorporating various strategies such as Counterfactual Reasoning Learning, Coherent Constrain Reasoning, and Causal Reasoning Learning. The paper highlights the importance of considering static, dynamic, and social biases in fake news detection, as these biases can impact the accuracy and generalization ability of detection models.

**Strengths:**

This paper presents an important perspective to the task of fake news detection: that of causality. It systematically constructs and learns the causal relationships between different modalities, as well as identifying several potential sources of bias. In addition to the novel perspective, the authors also considered the problem of highly believable fake news and proposed a relatively novel method (CCR) to resolve this issue. Experiments on two fake news datasets show strong performance, and ablation and sensitivity analyses are also complete.

**Limitations:**

I feel the design of the model is suboptimal, in the sense that, although the model is portrayed to be end-to-end, the three encoding methodologies respectively output different embedding outputs which are then analyzed in the transformer layer. Apart from this transformer connection, in no other place do the three submodules share their information. For example, one could imagine that the CRL module could be applied to text-images inputs as well, and the CFL module be applied to video-audio information. As it is presented, the different modules feel unnaturally fitted together.

In addition, I feel that not enough information has been provided with regards to the two causal learning modules (CRL and CFL). I feel that their respective learning strategies are slightly unconventional (e.g., incorporating the NDE terms into the loss function), but perhaps the authors can clarify.

**Suitability:**

3

---

### Meta-Review · Area_Chair_dtd9 · 2024-07-03

**Recommendation:** Accept (Poster)
**Confidence:** 5

**Metareview:**

After rebuttal,  all reviewers are in agreement that the paper's strengths warrant acceptance, and the AC agrees.

---

### Meta-Review · Senior_Area_Chairs · 2024-07-10

**Recommendation:** Accept (Poster)
**Confidence:** 5

**Metareview:**

This paper received mixed ratings initially. After rebuttal, all the reviewers tend to accept the paper. SAC and AC agree with reviewers and recommend acceptance of the paper.